# Insights of RKIP-Derived Suppression of Prostate Cancer

**DOI:** 10.3390/cancers13246388

**Published:** 2021-12-20

**Authors:** Ying Dong, Xiaozeng Lin, Anil Kapoor, Yan Gu, Hui Xu, Pierre Major, Damu Tang

**Affiliations:** 1Department of Surgery, McMaster University, Hamilton, ON L8S 4K1, Canada; dongy87@mcmaster.ca (Y.D.); linx36@mcmaster.ca (X.L.); akapoor@mcmaster.ca (A.K.); guy3@mcmaster.ca (Y.G.); 2Urological Cancer Center for Research and Innovation (UCCRI), St Joseph’s Hospital, Hamilton, ON L8N 4A6, Canada; 3The Research Institute of St Joe’s Hamilton, St Joseph’s Hospital, Hamilton, ON L8N 4A6, Canada; 4The Division of Nephrology, Xiangya Hospital of the Central South University, Changsha 410008, China; xuhuiye@csu.edu.cn; 5Department of Oncology, McMaster University, Hamilton, ON L8S 4L8, Canada; majorp@hhsc.ca

**Keywords:** prostate cancer, metastasis, RKIP, signaling events

## Abstract

**Simple Summary:**

Despite an intensive research effort in the past few decades, prostate cancer (PC) remains a top cause of cancer death in men, particularly in the developed world. The major cause of fatality is the progression of local prostate cancer to metastasis disease. Treatment of patients with metastatic prostate cancer (mPC) is generally ineffective. Based on the discovery of mPC relying on androgen for growth, many patients with mPC show an initial response to the standard of care: androgen deprivation therapy (ADT). However, lethal castration resistant prostate cancers (CRPCs) commonly develop. It is widely accepted that intervention of metastatic progression of PC is a critical point of intervention to reduce PC death. Accumulative evidence reveals a role of RKIP in suppression of PC progression towards mPC. We will review current evidence and discuss the potential utilization of RKIP in preventing mPC progression.

**Abstract:**

Prostate cancer (PC) is a major cause of cancer death in men. The disease has a great disparity in prognosis. Although low grade PCs with Gleason scores ≤ 6 are indolent, high-risk PCs are likely to relapse and metastasize. The standard of care for metastatic PC (mPC) remains androgen deprivation therapy (ADT). Resistance commonly occurs in the form of castration resistant PC (CRPC). Despite decades of research efforts, CRPC remains lethal. Understanding of mechanisms underpinning metastatic progression represents the overarching challenge in PC research. This progression is regulated by complex mechanisms, including those regulating PC cell proliferation, epithelial–mesenchymal transition (EMT), and androgen receptor (AR) signaling. Among this PC metastatic network lies an intriguing suppressor of PC metastasis: the Raf kinase inhibitory protein (RKIP). Clinically, the RKIP protein is downregulated in PC, and showed further reduction in mPC. In xenograft mouse models for PC, RKIP inhibits metastasis. In vitro, RKIP reduces PC cell invasion and sensitizes PC cells to therapeutic treatments. Mechanistically, RKIP suppresses Raf-MEK-ERK activation and EMT, and modulates extracellular matrix. In return, Snail, NFκB, and the polycomb protein EZH2 contribute to inhibition of RKIP expression. In this review, we will thoroughly analyze RKIP’s tumor suppression actions in PC.

## 1. Introduction

In the developed world, prostate cancer (PC) is the most frequently diagnosed male malignancy and a major cause of cancer death in men [1]. The disease is initiated from prostate epithelial cells as high-grade prostatic intra-epithelial neoplasia (HGPIN) lesions that evolve to invasive prostate adenocarcinoma or PC which can progress to metastasis [2]. PCs are graded based on Gleason scores (GS) or GS-based World Health Organization (WHO) grading system which categorizes PC into WHO grade group 1–5 [3,4,5].

Prostate cancers are characterized with a high degree of disparity in terms of its prognosis. While tumors with GS ≤ 6 or WHO grade group 1 are generally indolent, others possess high-risk of progression. Local PCs are managed with watchful waiting (active surveillance) and curative therapies: radical prostatectomy (RP) or radiation therapy (RT) [6,7,8,9]. Approximately 30% of patients will experience disease relapse or biochemical recurrence (BCR) based on increases in serum prostate-specific antigen (PSA) [10]. BCR is defined with elevations of serum PSA > 0.2 ng/mL after RP or > 2 ng/mL above the nadir following RT [11]. Relapsed tumors elevated risks of metastasis; 24–34% of patients following BCR will develop metastatic PC (mPC) [12,13].

Built on the androgen-dependence nature of PC discovered in 1940s, current standard of care for mPCs remains androgen deprivation therapy (ADT) [14,15]. Despite remarkable initial response in more than 80% of patients with mPC, ADT is essentially a palliative care as metastatic castration-resistant PCs (mCRPCs) commonly develop [16,17]. Owing to extensive research efforts, multiple options are available to manage CRPCs, including taxane-based chemotherapy, anti-androgens targeted therapy involving either abiraterone or enzalutamide [17,18,19], and immunotherapy [20,21]. However, these therapies only offer modest survival benefits [17,22]. From this perspective, interventions targeting early-stage progressions of BCR and metastasis are more desirable than treating CRPC or mCRPC.

Metastasis contributes to more than 90% of cancer deaths [23,24], and is regulated by complex networks. Epithelial–mesenchymal transition (EMT) is critical in promoting metastasis; EMT increases cancer cell’s migratory and invasion capacities, which are essential properties in facilitating the establishment of cancer cells at the secondary organs from primary site [25,26]. EMT is a major contributor to cancer stem cells [27], including prostate cancer stem cells (PCSCs) [28]. PCSCs are a major source of PC metastasis [28]. Other processes contributing to PC metastasis include cell proliferation regulated by Raf-MEK-ERK and PI3K-AKT-mTOR pathways [29,30], the EZH2 polycomb protein [31,32,33], and NFκB [34,35]. Intriguingly, all these processes are connected to a suppressor of PC metastasis, Raf kinase inhibitor protein (RKIP) (Figure 1) [36,37]. In this review, we will discuss the evidence supporting RKIP-derived suppression of PC pathogenesis and metastasis; the associated mechanisms and limitations will be addressed. This review will update the status of RKIP as a metastasis suppressor of prostate cancer and suggests future directions.

To provide a comprehensive review, we systemically searched PubMed for literature under the term: “RKIP AND prostate cancer”. A total of 51 articles were retrieved. Following exclusion of retraction-related (*n* = 2), non-cancer (*n* = 1) and non-PC-related (*n* = 8), non-English (*n* = 1), non-relevant articles (*n* = 2), and reviews (*n* = 10); 27 publications are discussed.

## 2. The Molecular Basis for RKIP as a Tumor Suppressor

RKIP (or PEBP1) belongs to the phosphatidylethanolamine-binding protein (PEBP) family. The protein was initially purified as a soluble basic protein with the molecular mass of 23 kDa from bovine brain [38]. RKIP/PEBP1 is a highly conserved protein expressed in mammalians including human [39,40], monkey [41], rat [42], mouse [43], chicken, and sheep [44]; its homologues were also detected in other organisms [45], including *Drosophila melanogaster* [46], *Saccharomyces cerevisiae* [47], *Toxocara canis* [48], *Onchocerca volvulus* [49], *Plasmodium falciparum* (a malarial parasite) [50], as well as flowering plants: *Antirrhinum* [51] and *Arabidopsis* thaliana [52]. The expression of RKIP in such a wide range of species supports its essential roles in a multitude of critical biological processes.

One of the most well-characterized processes affected by RKIP is its inhibition of Raf1-mediated activation of the MEK-ERK pathway. RKIP binds to Raf1 and MEK1, and prevents Raf1 in activating MEK [53]. PKC phosphorylates RKIP at serine 153 (S153), causing its dissociation with Raf1 [54] (Figure 2; Raf1 signaling). In support of this knowledge, *RKIP1*^−/−^ mice display elevations in ERK1/2 activation in the suprachiasmatic nucleus (SCN) in response to light [55]; conversely, in transgenic mice expressing non-phosphorylated form of RKIP, light-induced ERK1/2 activation was reduced in SCN [55].

Phosphorylation of S153 not only results in RKIP dissociation with Raf-1, which elevates the actions of the Raf-1-MEK-ERK pathway, but also enhances G protein-coupled receptor (GPCR) signaling (Figure 2). The G protein-coupled receptor kinase 2 (GRK2) phosphorylates active GPCRs, causing their decoupling from G proteins and thus attenuating GPCR signaling [56] (Figure 2). S153 phosphorylated RKIP (RKIP(S153P)) binds GRK2, which prevents GRK2 from phosphorylating GPCR, stabilizes GPCR-G protein complex, and thus facilitates GPCR signaling [57] (Figure 2; GPCR signaling). This regulation is physiologically relevant. Cardiac-specific expression of wild type but not S153 phosphorylation deficient mutant (S153A) increases β-adrenergic receptor (βAR), a 7-transmembrane GPCR involved in cardiac contractility, leading to protect mice from heart failure induced by chronic pressure overload; conversely, RKIP knockout mice are associated with exacerbation of pressure overload-induced heart failure [58,59].

The Raf-MEK-ERK axis is widely involved in oncogenesis [60,61,62] together with G protein-coupled receptors [63,64]. RKIP status alterations thus coordinate multifaceted oncogenic actions, including Raf-MEK-ERK, GPCR, and immune actions. Tumorigenesis is intimately connected with immunity [65]. Evidence supports a role of RKIP in regulating immunoreactions. Binding of IgE to its high affinity receptor FcɛRI activates mast cells, leading to production of proinflammatory cytokines and allergic asthma [66]. RKIP reduces IgE-FcɛRI-induced mast cell activation via inhibition of PI3K activation, leading to decreases in the production of proinflammatory cytokines [67]. *RKIP*^−/−^ mice sensitizes mast cell activation along with activation of the PI3K and AKT [67] (Figure 3A). Inflammation is tightly associated with cancer initiation and progression [68]. Intriguingly, in *RKIP*^−/−^ mice, T cell receptor of CD8+ cells specific to staphylococcal enterotoxin A is defective in its downstream signaling, leading to a decrease in INF-γ production [69] (Figure 3B). INF-γ and TNFα produced by CD8+ cytotoxic T lymphocyte (CTL) contribute to CTL-derived cytotoxicity to cancer cells [70]. RKIP also plays a role in innate immune response. In *RKIP*^−/−^ mice, the production of IFN-β, IL-6, and TNFα by macrophage’s Toll-like receptor 3 (TLR3) in response to polyinosinic:polycytidylic acid are significantly reduced [71] (Figure 3C). Similarly, vesicular stomatitis virus (VSV) and herpes simplex virus (HSV) induce type I interferon production to a significantly reduced level in *RKIP*^−/−^ mice [72]. In facilitation of these innate immune responses, phosphorylation of RKIP at S109 is required; the phosphorylation leads to binding of RKIP with TANK-binding kinase 1 (TBK1), which triggers the aforementioned innate immunity (Figure 3C) [71,72]. Macrophages, type I interferons, IFN-β, and TNFα can suppress cancer, while their impacts on tumorigenesis can be complex [73,74,75,76]. In line with the concept of RKIP contributing to tumor suppression via modulating immune responses, RKIP inhibits NFκB activation via downregulating IκB kinase (IKK) [77]; NFκB is a critical transcription factor that regulates immunity [78,79] and promotes tumorigenesis [80].

RKIP stabilizes glycogen synthase kinase 3-β (GSK3β) via binding to GSK3β, in turn preventing GSK3β phosphorylation at T390 by p38 MAPK [81], an event leading to GSK3β degradation [82]. In comparison to wild type mice, GSK3β expression is significantly reduced in the prostate epithelial cells of *RKIP*^−/−^ mice [81], supporting the regulation being physiologically relevant. GSK3β possesses tumor suppressing function via inhibiting the Wnt/β-catenin pathway [83], EMT, and cyclin D1 expression [81].

Accumulative evidence reveals RKIP’s impact in multiple processes towards suppression of tumorigenesis. Recent development using transgenic RKIP mice supports these RKIP-associated processes being physiologically relevant.

## 3. RKIP as a Tumor Suppression of Prostate Cancer (PC)

Consistent with RIKP’s potential role in molecular events relevant to tumor suppression, RKIP downregulations, and RKIP’s associations with cancer progression, RKIP-derived tumor suppressive activities have been reported in multiple cancer types, including urogenital cancers (bladder cancer [84], clear cell renal cell carcinoma [85,86,87], and PC [88,89]), breast cancer [90], pancreatic cancer [91], hepatoma [92], non-small cell lung cancer [93], gastric cancer [94], and others [95].

### 3.1. RKIP-Mediated Suppression of PC Tumorigenesis and Metastasis

#### 3.1.1. Facilitation of PC Initiation and Metastasis via Downregulation of RKIP at the Protein Level

The first evidence for RKIP as a metastatic suppressor of PC started with the identification of RKIP downregulation in LNCaP cells-derived metastatic C4-2B cells compared to their parental cells [96]; functionally, downregulation of RKIP elevated metastatic potential of C4-2B cells [88]. Specifically, restoration of RKIP expression in C4-2B cells to a comparable level in LNCaP cells reduced C4-2B cells’ invasion ability in vitro and the cells’ ability to produce lung metastasis in an orthotopic PC model without affecting its ability in forming primary tumors [88]. Downregulation of RKIP at the protein level was observed following PC progression from low grade (low Gleason score) to high grade and the downregulation was particularly evident in metastatic PCs (*n* = 22) [88]. In comparison to LNCaP cells, C4-2B cells displayed an increase in ERK activation, and inhibition of ERK activation with PD098059 decreased C4-2B cell invasion capacity in vitro.

Further analysis of RKIP expression in a tissue microarray containing non-tumor prostate tissues (*n* = 57), primary PCs (*n* = 79), and metastatic CRPC (*n* = 55), RKIP downregulation was detected in 48% of primary (or local) PCs and 89% of mCRPCs respectively [89]. RKIP downregulation in primary PCs stratifies the risk of PC recurrence (biochemical recurrence) following surgery and remains an independent risk factor of relapse after adjusting for Gleason score, maximal tumor diameter, pathological stage, surgical margin status, digital rectal examination, PSA, and gland weight [89].

The RKIP downregulations in primary PC compared to non-cancerous prostate tissues and its further downregulation in mPC vs. primary PC also occurred following PC progression in TRAMP mice [97]. While systemic knockout of RKIP had minimal impact on mouse health [98], RKIP deficiency significantly enhanced PC formation and metastasis in TRAMP mice [97]. Collectively, clinical and transgenic mouse (functional) studies support RKIP’s action in suppressing PC tumorigenesis and metastasis. However, whether decreases in PC metastasis in *RKIP*^−/−^*;TRMAP* mice were a direct result of reductions in primary PC formation requires additional investigations. Major limitations of the above studies include lack of analyses of RKIP expression at the mRNA level and utilization of more targeted transgenic model such as prostate-specific *RKIP*^−/−^ mice.

#### 3.1.2. No Apparent Reduction of RKIP mRNA Expression Following PC Pathogenesis

The number of publications related to RKIP reductions following PC evolution remains limited. This might be in part attributed to the highly heterogenous nature of PC and challenges in studying RKIP downregulation at the protein level. With advances in DNA sequencing (Next-generation sequencing), an ever-increasing number of cancer genetics and gene expression data have been accumulated and made available. Using TCGA RNA-seq data on prostate tissue (*n* = 52) and PC samples (*n* = 497) available from the UALCAN platform (ualcan.path.uab.edu/home, accessed on 6 October 2021) [99], we did not find apparent downregulation of RKIP in PC, high grade PCs, and lymph node metastasis (Figure 4A–C). Similar observations could be obtained using the GEPIA2 platform [100] with more prostate tissues (*n* = 152) (Figure 4D). Furthermore, using the Sawyers microarray dataset [101] within the R2: Genomics Analysis and Visualization platform (http://r2.amc.nl, accessed on 3 December 2021), RKIP mRNA was not significantly expressed at reduced levels in mPCs compared to primary PCs (Figure 2E). This strongly suggests the RKIP downregulation observed in PC occurs at least in part at post-mRNA levels. Future research should explore these mechanisms.

### 3.2. Regulation of RKIP Expression in PC Cells

#### 3.2.1. RKIP as a Target of Androgen Receptor (AR)

Androgen signaling plays dominant roles in PC initiation, progression, and CRPC development [28,102,103]. This knowledge implies a relationship between AR and RKIP, which is supported by experimental evidence. In immortalized and non-tumorigenic human prostate epithelial RWPE-1 cells [104], dihydrotestosterone (DHT) upregulated RKIP transcription which was blocked by antiandrogen bicalutamide [105]. Androgen response element (ARF), an AR-binding DNA motif, was identified in the RKIP promoter region between nucleotide −571 and −548. A RKIP promoter fragment (−2206 to −26) encompassing this region mediated reporter expression in response to DHT in RWPE-1 cells [105]. Castration of C57BL/6 mice significantly reduced RKIP mRNA expression in prostate, providing a physiological relevance of RKIP as an AR target.

The relationship between AR and RKIP in PC pathogenesis and progression might be complex. AR signaling is required for prostate development, evident by the lack of prostate in *AR*^−/−^ mice [106] and humans with AR mutations being completely insensitive to androgen [107,108]. In adults, AR function is essential for the maintenance of luminal secretory epithelial cells [108]. Alterations in AR signaling are the major mechanism underlying all aspects of PC pathogenesis, including CRPC development [28,102,103,108]. These alterations may lead to changes in AR targets in normal prostate epithelial cells, prostate cancer, and following PC progression. In this regard, RKIP expression is reversely correlated with PSA expression, a typical AR target [109], in PCs [110]; serum PSA is the only clinically used biomarker assessing PC relapse and disease severity [10,111,112]. The selective expression of PSA over RKIP in PC is consistent with the concept that loss of AR targets with tumor suppression activities contributes to AR-promoted PC initiation and progression. This concept is consistent with AR’s complex actions in PC: inhibiting c-Met and AKT activation (both promoting PC progression) in PC [113,114,115] and attenuation of PC3 cell proliferation [116]. It thus would be interesting to determine the impact on PC by enforcing RKIP as an AR target in PC, for instance placing RKIP under the control of PSA promoter.

#### 3.2.2. Mutual Regulation of RKIP and the EMT Machinery

As a suppressor of metastasis, RKIP preferentially inhibits LNCaP, C4-2B, and PC-3M cells invasion but not proliferation in vitro [88,117]. This is not unique in PC cells; similar observations were also reported in clear cell renal cell carcinoma A498 cells [86]. EMT plays a major role in enhancing cancer cell invasion capacity in vitro and metastasis in vivo [25,26]. In a HEK-293 cells-based study, RKIP downregulates Snail and Slug expression via stabilization of GSK3β [81]. Both Snail and Slug are major transcription factors of EMT [118,119]. Snail increases LNCaP cell migration in vitro through facilitating degradation of the SPOP tumor suppressor [120]; EMT plays a major role in the generation of prostate cancer stem cells which contribute to PC metastasis and CRPC development [17,28]. In accordance with this evidence, Snail inhibits RKIP expression in metastatic and AR-negative PC3 and DU145 PC cells [121]; this inhibition occurs at the transcription level through an E-box located in the RKIP promoter [121]. A reverse correlation between Snail and RKIP mRNA expressions was observed in primary PCs [121]. The connection between Snail and RKIP likely has a functional impact on PC. Downregulation of Snail sensitized DU145 PC cells to TRAIL- and CDDP (Cisplatin)-induced apoptosis via upregulation of RKIP and knockdown of RKIP reversed the sensitization [122].

Evidence presented above suggests a mutual inhibition between RKIP and Snail. While RKIP can downregulate Snail in HEK-499 (a derivative of HEK-293 cells) cells [81], whether this occurs in PC cells remains to be demonstrated. Nonetheless, RKIP may inhibits PC cell invasion and migration independent of the core transcriptional factors of EMT. RKIP inhibits PC-3M cell migration and invasion in vitro via modulating extracellular matrix by downregulating MMP-2 and MMP-9 (matrix metalloproteinases) [117].

#### 3.2.3. Non-Coding RNA-Mediated Downregulation of RKIP in PC Cells

It is an emerging concept that long non-coding RNAs (lncRNAs) sponge microRNAs (miRNAs or miRs) and thus facilitate mRNA expression [123,124]. In this regard, miR-543 was reported to downregulate RKIP in PC and thus promote PC cell proliferation and EMT [125]. In LNCaP and C4-2B cells, downregulation of RKIP and upregulation of miR-543 occur concurrently in C4-2B cells [125]. The presence of miR-543 target sequence was detected in the 3′UTR (untranslated region) of RKIP mRNA and expression of miR-543 but not its negative control reduces RKIP mRNA expression in LNCaP cells, supporting RKIP as a direct target of miR-543 [125]. Ectopic expression of miR-543 enhances LNCaP cell proliferation, invasion, and xenograft formation along with evidence of EMT; conversely, these events were reduced upon knockdown of miR-543 in C4-2B cells [125], supporting a likely functional impact of miR-543 in inhibition of RKIP. Additionally, a reverse correlation between miR-543 and RKIP expression was demonstrated in a PC cohort consisting of *n* = 28 local tumors and *n* = 14 metastatic PCs [125].

The same group also reported a regulation between lncRNA XIST and RKIP expression in PC. Specifically, XIST sustains RKIP expression through binding to miR-23a. In clinical samples, concurrent downregulation of XIST and RKIP occurs in primary PC vs. normal prostate and mPC vs. primary PCs [126]. Ectopic expression of XIST in DU145 cells increases RKIP expression, decreases cell proliferation, and attenuates xenograft formation [126]. In a reverse manner, knockdown of XIST in LNCaP cells, which express a high level of endogenous XIST, downregulates RKIP along with an enhancement in cell proliferation [126]. In support of this investigation, miR-23c, a close relative of miR-23a, was suggested to target RKIP [127].

MiRNA likely has many targeted genes or mRNAs; for instance, miR-130b has approximately 600 target genes [128]. MiRNA thus affects complex network alterations. In this regard, miR-543 and XIST-miR-23a likely impact PC progression with multiplex mechanisms in addition to regulating RKIP expression. This inference is supported by miR-543-derived tumor-promotion in lung cancer [129,130] and tumor-inhibition in colorectal and cervical cancers [131,132]. Even in PC, downregulation of miR-543 was reported in primary PC with bone metastasis (*n* = 20) compared to primary PCs without bone metastasis (*n* = 15), and in bone mPCs compared to the paired primary PCs; the reported target of miR-543 in this investigation was endothelial nitric oxidase (eNOS) [133]. In PC3 cells, miR-543 downregulates eNOS and inhibits PC3 migration [133]. On the other hand, miR-543 can also enhance PC cell oncogenic properties via stimulating the AKT/mTOR pathway [134] and enhancing prostate cancer stem cell traits [135]. While modulations of miR-543 [125] and XIST [126] leading to RKIP downregulation were associated with increases in PC cell proliferation, direct downregulation of RKIP did not affect PC cell proliferation [88,117]. Collectively, miR-543 and XIST likely impact PC cell oncogenic properties via multiple targets, including RKIP.

### 3.3. RKIP-Derived Sensitization of PC Cells to Treatment In Vitro

Accumulative investigations present a consistent message for RKIP as tumor suppressor and/or metastatic suppressor of PC. This concept is further supported by RKIP’s action in sensitization of PC cells to multiple cytotoxic treatments in vitro.

RKIP contributes to DU145 cell response to TRAIL- and cisplatin-induced apoptosis; this sensitization was reversed upon inhibition of RKIP expression by Snail [122]. Similarly, nitric oxide (NO) inhibits EMT in PC3 and DU145 cells via RKIP upregulation and Snail downregulation. RKIP upregulation in this setting makes a major contribution to EMT inhibition caused by Snail downregulation [136].

RKIP plays a role in PC cell’s sensitivity to photodynamic therapy (PDT) in response to NO levels produced during PDT. In PC3 cells treated with PDT, optimized treatment condition led to the production of a high level of NO, which inhibits NFκB and YY1 (Yin Yang 1) transcription factor. As a result, RKIP is upregulated and resulted in cytotoxicity [137,138]. Sub-optimal PDT treatment produces low levels of NO, a condition that activates NFκB-mediated YY1 expression. YY1 subsequently inhibits RKIP, contributing to EMT and the activation of PI3/AKT [137,138]. YY1 promotes PC via EMT development and contributes to therapy resistance [139]. In addition to the NFκB-YY1-RKIP connection, high and low NO levels can also modulate EMT and drug resistance in PC3 cells via NFκB-RKIP-GSK3β-NRF2, where high NO inhibits NFκB, leading to RKIP upregulation, GSK3β stabilization and NRF2 downregulation along with inhibition of EMT and sensitization to drug treatment. Low NO produces the opposite actions [140].

Evidence suggests a contribution of RKIP to genotoxic agent 9-nitrocamptothecin (9NC)-induced apoptosis in PC cells [141]. 9NC triggers DNA damage response and apoptosis along with RKIP upregulation in DU145 cells but not in 9NC-resistant RC1 cells which were derived from DU145 cells. Sensitivity of PC cells to 9NC-induced apoptosis was reduced or increases with RKIP downregulation and overexpression respectively [141]. It was indicated that NFκB contributed to RKIP expression alteration and RKIP sensitized PC cells to DNA damage-induced apoptosis [141]. This investigation was supported by a report that radiation upregulated RKIP in C4-2B cells [142]. RKIP overexpression and knockdown sensitized and reduced C4-2B cell apoptosis in vitro in response to radiation [142]. In mice bearing tumors produced by C4-2B cells with RKIP knockdown, radiation was substantially less effective in inhibiting tumor growth compared to tumors generated by wild type C4-2B cells [142]. Collectively, evidence supports RKIP playing a role in PC cell response to genotoxic treatment. As radiation is clinically used in treating PC, whether RKIP is a major contributor to the therapy response needs further investigation.

Docetaxel is commonly used in treating CRPC [143,144]. It was observed that in PC3 cells, a PC cell line that does not require androgen for survival, RKIP overexpression sensitized the cells to docetaxel-induced inhibition of cell proliferation [145]. While this report shows a potential for clinical implication, more work is required to further explore this potential.

## 4. Utilization of Additional Mechanisms in RKIP-Derived Inhibition of PC

RKIP is involved in numerous signaling events under different settings, including those of well established: the Raf-MEK-ERK, GPCRs, NFκB, Snail, and GSK3β. Most of these regulations are physiologically relevant, evident by their occurrence in RKIP transgenic mice (see Section 2). Except PKC-RKIP(S153P)-GPCR, RKIP clearly utilizes the above connections in suppressing PC. In addition to these well-established signaling pathways, RKIP may also explore other processes critical to PC.

### 4.1. A Potential Interplay between Two PC Metastasis Suppressors: RKIP and Annexin A7

Annexin A7 (ANX7; Annexin VII) is encoded by *ANXA7* gene located at 10q22.2 (https://www.genecards.org/cgi-bin/carddisp.pl?gene=ANXA7, accessed on 1 October 2021). ANX7 binds Ca2+ and displays GTPase activity [146,147]. The impact of ANX7 on tumorigenesis appears to be tumor-type dependent [148]. Nonetheless, evidence supports ANX7 as a metastatic suppressor of PC. Significant downregulation of ANX7 occurs in mPC and CRPC [149,150]. ANX7^+/−^ mice (23%) spontaneously develop tumors in multiple organs, including prostate [151]. Ectopic expression of ANX7 resulted in inhibition of proliferation and cytotoxicity in LNCaP, PC3, and DU145 cells [150,152]. Consistent with this knowledge, it was recently reported that ANX7 suppresses PC3 cell metastatic properties via activation of AMPK, resulting in decreasing mTOR and STAT3 actions [153]. Surprisingly, RKIP binds ANX7 and reduces ANX7-mediated PC suppression [153]. This study indicates that RKIP and ANX7 suppress PC metastasis alone. While this shows an intriguing interplay, its molecular insights and clinical relevance require further investigations.

### 4.2. EZH2-Derived Downregulation of RKIP Transcription in PC

EZH2 is the enzymatic subunit of the Polycomb Repressive complex 2 and facilitates the trimethylation of histone H3 at lysine 27 (H3K27me3) [154], a typical modification leading to transcriptional suppression [155,156]. This action of EZH2 underlines its promotion of cancer progression and metastasis in multiple tumor types [157,158,159]. EZH2 upregulation occurs in prostate cancer stem cells (PCSCs) and plays a critical role in PCSC growth [160]. PCSCs makes major contributions to PC metastasis and CRPC development [17,28]. Evidence supports RKIP being a critical target silenced by EZH2 in facilitating PC metastasis [161]. A reverse pattern of RKIP and EZH2 expression in primary PC and mPC was observed, which correlated with PC progression (progression free survival) [161]. EZH2 mediates downregulation of RKIP in LNCaP and DU145 cells; this suppression was resulted via recruiting of EZH2 to the RKIP promoter, a process regulated by Snail [161]. In view of the major role of EZH2 in promoting PC progression and metastasis, the involvement of RKIP downregulation in these processes supports RKIP as a metastatic suppressor of PC.

### 4.3. Immune Alterations Associated with RKIP in PC

Tumorigenesis and cancer progression are intimately linked with immune alterations [65]; tumor-associated immune cells play critical roles in tumor initiation and progression [162,163]. Recent research using RKIP transgenic mice revealed important roles of RKIP in adaptive and innate immune reactions in non-cancer settings (see Section 2 and Figure 3). RKIP involvement in tumorigenesis in the aspect of immune alterations has not been investigated. In this section, we will explore whether RKIP expression is associated with immune alterations relevant to PC.

Immune evasion is essential for cancer progression. PD-L1 is an immune checkpoint protein and plays a major role in PC to escape immune attack [164]. Using TISIDE (an integrated repository portal for tumor-immune system interaction) platform [165], we observed a significant reverse correlation (Spearman r = −0.503, *p* < 2.2 × 10^−16^) between RKIP and CD274 (PD-L1) expression in PC (*n* = 498, TCGA PC dataset) (Figure 5). This strong negative correlation also occurs in bladder cancer, breast cancer, lung adenocarcinoma, rectal cancer, and thyroid carcinoma (Table 1). Similarly, PD-L2 (PDCD1LG2: programmed cell death 1 ligand 2) is an immune checkpoint that contributes to immunosuppressive microenvironment for cancer [166]; its expression is reversely correlated with RKIP expression in PC (Spearman r = −0.384, *p* < 2.2 × 10^−16^) (Figure 5). Negative correlations between RKIP and PD-L2 are present in other cancer types (Table 1; note: this is for illustration only and does not intend to be inclusive, i.e., some cancer types with significant negative correlation are not included). RKIP expression in PC also shows strong negative correlations with BTLA (B and T lymphocyte attenuator), CD96, TIGIT (T cell immunoreceptor with immunoglobulin and ITIM domain), and CSF1R (Figure 5). BTLA, CD96 and TIGIT are immune checkpoints that play major roles in the formation of tumor-permissive microenvironment [167,168]. CSF1R contribute to cancer-associated immunosuppressive microenvironment via regulating TAM (tumor-associated macrophage) [169]. Furthermore, RKIP expression is also negatively associated with PD-L1 (Spearman r = −0.28, *p* = 4.17 × 10^−5^), PD-L2 (Spearman r = −0.283, *p* = 3.489 × 10^−5^), BTLA (Spearman r = −0.362, *p* = 4.17 × 10^−8^), CD96 (Spearman r = −0.273, *p* = 6.431 × 10^−5^), TIGIT (Spearman r = −0.200, *p* = 0.003687), and CSF1R (Spearman r = −0.261, *p* = 0.0001371) in metastatic PCs (*n* = 208) in the SU2C dataset [170] within cBioPortal [171,172]. Collectively, evidence suggests a role of RKIP in facilitating immune attack on PC cells, which may contribute to its suppression of PC metastasis.

The above notion is also in accordance with a strong negative correlation between RKIP expression and the Treg (T regulatory) cell population (Figure 5). Treg cells suppress T cells activation via downregulation of CD80/86 in antigen-presenting dendritic cells [173], and are a major contributor to immunosuppressive microenvironment.

## 5. Perspectives

Since the identification of RKIP as a candidate suppressor of PC metastasis by Fu et al. in 2003 [88], accumulative research reinforces RKIP as a tumor suppressor and metastatic suppressor not only in PC but also in other cancer types [95]. The notion of RKIP as suppressor of metastasis is supported by (1) its actions in inhibiting cancer cell migration and invasion but not proliferation in vitro [86,88,117], (2) its ability in suppressing PC metastasis in xenograft models [88] and transgenic mice [97], as well as (3) its significant downregulation in mPC [88].

Downregulation of RIKIP commonly occurs in multiple cancer types including PC, clear cell renal cell carcinoma, and others [86,88,95]. However, mechanisms leading to RKIP downregulations in PC and other cancer types remain unclear. Evidence supports the downregulation happening at the protein level [88] but not at the mRNA level (Figure 4). While numerous mechanisms (Snail, EZH2, NFκB, YY1) can inhibit RKIP transcription in PC cells in vitro (see above sections for details), RKIP mRNA expression is not evidently reduced in primary PCs, following increase in PC severity (Gleason score), lymph node metastases, and distant metastases (Figure 4). The absence of insights on RKIP downregulations following PC pathogenesis presents a major challenge to further understand RKIP-derived suppression of PC metastasis and the exploration of this knowledge for clinical applications. For instance, will it be possible to prevent RKIP downregulation to attenuate PC metastasis? In view of RKIP expression in multiple tissues [174], it may be important to investigate the impact of RKIP on PC using prostate specific transgenic mice (knockout and overexpression) together with typical oncogenic signals like PTEN deficiency.

It remains unclear whether RKIP plays a role in CRPC development. Nonetheless, this potential should be explored. This possibility is supported by a critical role of RKIP silencing in EZH2-derived promotion of PC [161]. EZH2 suppresses AR expression, facilitates the reprogramming of PC to PCSCs, and contributes to AR-independent growth of PC [175]. Persistant AR signaling is a major mechanism leading to CRPC [17,176,177]. Considering RKIP being an AR target in normal prostate epithelial cells [105], it remains a possibility that the selective removal of RKIP from the AR target gene list facilitates CRPC development. With prostate-specific RKIP transgenic mice, RKIP’s involvement in resistance to ADT should be investigated.

Our brief in silico analysis indicates a role of RKIP in facilitating immune attack on PC (see Section 4.3.). Despite the lack of literature supporting this scenario, critical roles of RKIP in immune responses in non-cancer investigations (see Section 2 and Figure 3) supports our suggestion. Considering the observation for non-apparent reductions of RKIP mRNA expression in PC (Figure 4), it is an appealing possibility that RKIP facilitates a non-permissive microenvironment for PC via regulating tumor microenvironment.

An important aspect of RKIP biology relates to its shifting status: S153 phosphorylation by PKC can switch its tumor suppressive actions to tumor-stimulation functions via activating GPCR [174,178] (Figure 2). This phosphorylation is facilitated by residue proline 74 (P74) within the ligand-binding pocket of RKIP; both the P74 and the ligand-binding pocket are conserved within the PEBP family [178,179,180]. Mutation P74L enhances RKIP phosphorylation at S153 and increases the activity of Raf1/ERK signaling [179]. The equivalent mutation has a physiological consequence in tomato plant to switch developments [181]. It remains a possibility that phosphorylation of RKIP at S153 is inhibited to preserve its actions in suppressing PC metastasis.

## 6. Conclusions

The research activities in the past 20 years collectively demonstrated RKIP as a tumor suppressor of PC tumorigenesis and metastasis. This knowledge is supported by (1) functional evidence derived from in vitro, in vivo (xenograft and transgenic mouse models), and clinical studies as well as (2) mechanistic pathways contributing to RKIP-derived suppression of PC. Future research should explore the functionality and underlying mechanisms for RKIP mediated suppression of PC using more refined transgenic models, including mice with prostate-specific expression of RKIP and its mutants. The latter may help to define the regulations relevant to RKIP tumor suppressive actions; this is important, as RKIP can be switched to promote tumorigenesis following its phosphorylation at S153 (Figure 2). Additionally, mechanisms leading to RKIP downregulation in PC and RKIP’s involvement in other aspects of PC progression should also be investigated (see Section 5 for details).

## Figures and Tables

**Figure 1 cancers-13-06388-f001:**
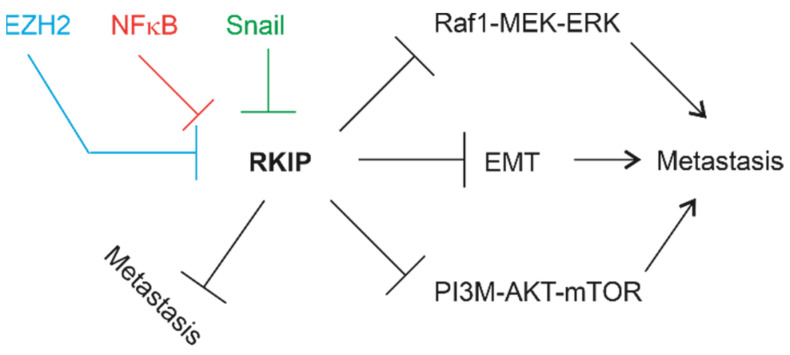
Factors and pathways that promote PC metastasis and display connections with RKIP.

**Figure 2 cancers-13-06388-f002:**
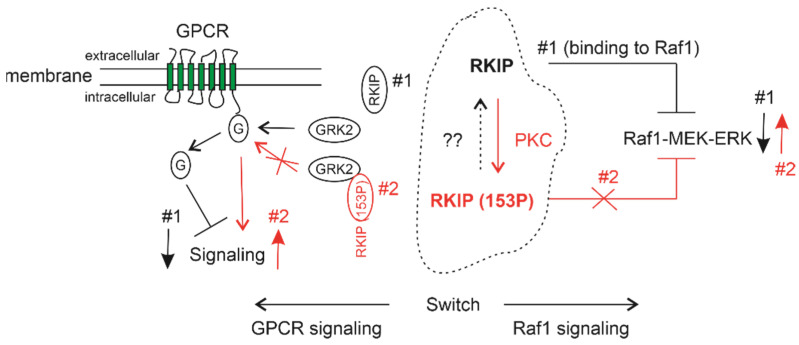
RKIP is a switch in regulating the Raf1-MEK-ERK and GPCR signaling. The dotted regions shows that RKIP can be phosphorylated at serine 153 (S153) by PKC and reversed to non-phosphorylated status by unknown factors marked with “??”. At non-phosphorylated status (#1), RKIP binds Raf1 and inhibits Raf1-MEK-ERK signaling. Non-phosphorylated RKIP does not associated with GRK2, which enables GRK2 to phosphorylate active GPCR, leading to G proteins to dissociate from GPCR and thereby inhibiting GPCR signaling. RKIP(S153P) is unable to bind or dissociate from Raf1 (#2), resulting in activation of the Raf1-MEK-ERK signaling. RKIP(S153P) binds to GRK2 and prevents GRK2 from uncoupling GPCR from G protein, in turn stabilizes the GPCR-G protein complex and promotes GPCR signaling.

**Figure 3 cancers-13-06388-f003:**
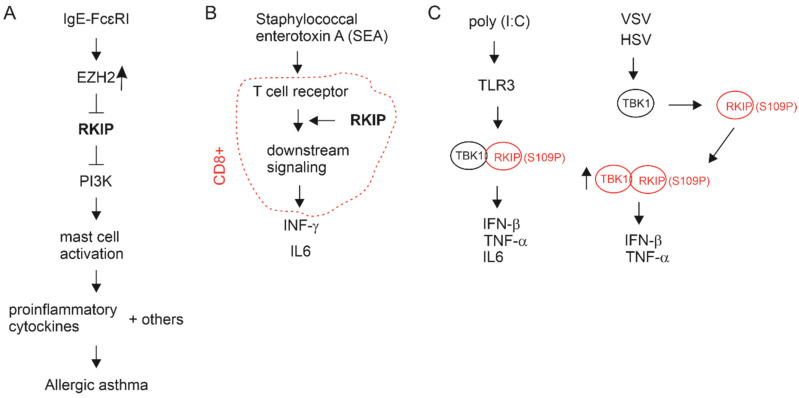
RKIP regulates immune reactions in vivo. The information presented here was based on literature involving RKIP transgenic mice. (**A**) Ligation of IgE to FcɛRI on mast cell surface leads to upregulation of EZH2 that inhibits RKIP transcription. The action reduces RKIP-mediated inhibition of PI3K which promotes mast cell activation, leading to the production of proinflammatory cytokines and other factors, followed by allergic asthma [67]. (**B**) RKIP facilitates T cell receptor downstream signaling events, resulting in the production of INF-γ and IL-6 in CD8+ T cells treated with SEA [69]. (**C**) Polyinosinic:polycytidylic [poly(I:C)] activates TLR3 (Toll-like receptor 3), leading to phosphorylation of RKIP at S109 (S109P) which binds to and activates TBK1 (TANK-binding kinase 1); TBK1 promotes the production INF-γ, TNF-α, and IL-6 [71]. VSV (vesicular stomatitis virus) and HSV (herpes simplex virus) infections induce phosphorylation of RKIP at S109 by TBK1, which then binds to TBK1, facilitates TBK1 autophosphorylation, and enhances TBK1 activity which in turn contributes to INF-γ and TNF-α production [72].

**Figure 4 cancers-13-06388-f004:**
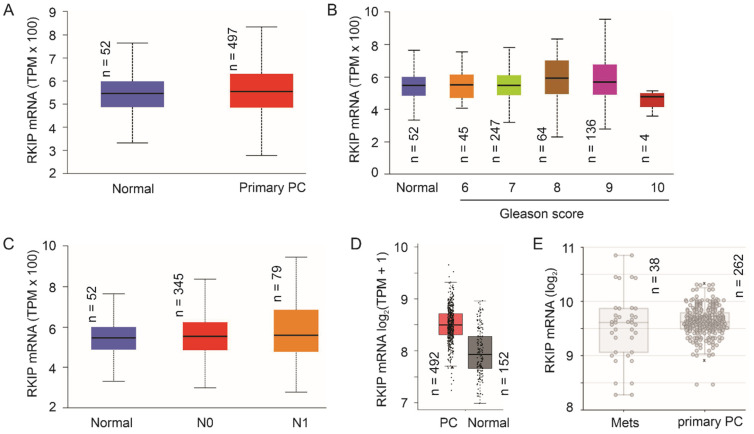
No apparent downregulation of RKIP mRNA expression in PC. (**A**–**C**) Images were generated using the UALCAN platform. (**D**) The graph was produced using the GEPIA2 platform. TPM: transcript per million. (**E**) RKIP mRNA expression in distant metastases (Mets) and primary PCs. The image was generated using the Sawyers dataset within the R2: Genomics Analysis and Visualization platform.

**Figure 5 cancers-13-06388-f005:**
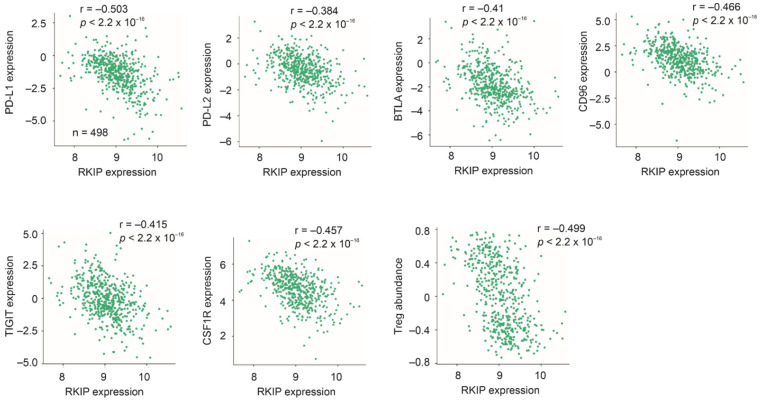
Negative associations of RKIP expression with the indicated immune checkpoints, CSF1R, and Treg (T regulatory cells). Images were produced using TISIDE. Spearman r (correlation coefficient) for the individual correlations are indicated. TCGA dataset (*n* = 498) was used.

**Table 1 cancers-13-06388-t001:** Negative correlations between RKIP and PD-L1 and PD-L2 in cancers ^a^.

Cancer Type	Population (*n*)	Spearman r	*p* Value
Bladder uroghelial carcinoma	408	−0.432 ^b^; −0.469 ^c^	<2.2 × 10^−16^; <2.2 × 10^−16^
Breast cabcer	1100	−0.41; −0.4	<2.2 × 10^−16^; <2.2 × 10^−16^
Lung adenocarcinoma	517	−0.428; −0.325	<2.2 × 10^−16^; 4.47 × 10^−14^
Rectum adenocarcinoma	167	−0.414; −0.36	3.66 × 10^−8^; 2.09 × 10^−6^
Thyroid carcinoma	509	−0.55; −0.415	<2.2 × 10^−16^; <2.2 × 10^−16^

^a^: analyses were performed using TISIDE; ^b^: correlation with PD-L1; ^c^: correlation with PD-L2.

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
