# Peer review of "Insights of RKIP-Derived Suppression of Prostate Cancer"

_cancers, 2021, doi:10.3390/cancers13246388_

Round 1

Reviewer 1 Report

The work of Dong et al., entitled “Insights of RKIP-derived suppression of prostate cancer” provides an up-to-date review regarding potential 21 utilization of RKIP in preventing mPC progression. I believe this manuscript is an interesting paper suitable to be published in the journal.

The review is divided into different parts that describe the molecular mechanisms underlying the in regulation of RKIP gene expression, the molecular basis for RKIP as a tumor suppressor and its specific role in metastatic prostate cancer. The manuscript is well structured, easy to read and with some interesting data summarized by the authors.

I recommend the manuscript for publication.

Author Response

We appreciate the reviewer’s positive comments.

Reviewer 2 Report

correct discussion

Author Response

We appreciate the reviewer's comment: “correct discussion”

Authors' response – We view this comment as being related to section 5 “Conclusions and perspectives”. This section in the last submission served two purposes: conclusion and suggestion for future works. This setting might have resulted in neither purpose being well served, a point also raised by Reviewer #3. In this revision, we separated this section into two sections: section 5: Perspectives and section 6: Conclusions. We trust the revised setting has solved the structural deficiency.

Reviewer 3 Report

In this review the authors consider the role of RKIP in prostate cancer and metastasis.  There are a number of issues with this review.

1) The English grammar needs extensive editing by a native English speaker qualified in the field.  There are numerous errors, some of which are meaningful.  For example, the authors use 'master cells'  instead of 'mast cells' on lines 131, 154, and 155.  On line 151 they use 'transfection factor' instead of 'transcription factor.'

2) Figure 4 indicates that there is no change in RKIP transcript levels for localized prostate cancers.  It would be helpful if the authors include a similar panel and analysis for metastatic prostate cancer using SU2C or other similar datasets.  Also, are there differences in patients with mutations in KRAS?

3) There seem to be some conflicts in the literature regarding whether changes in RKIP affect cellular proliferation.  Some papers say that it does and others that it does not.  The authors should comment as to why that might be, and try to resolve this or offer strengths and weaknesses of the different studies.

4) The authors discuss miRNAs and lncRNAs extensively, and the fact that the mRNA of RKIP doesn't change but the protein does, and yet they focus on the effects of miRNAs on transcript levels and not on potential translation efffects.

5) The first paragraph on page 8 has a long discussion of miR-453.  It is not clear why this is relevant entirely.  Also, it is also referred to as miR-543.  It is not clear if this is a typo or two different miRNAs.

6) The sentence on line 342 should have a reference at the end.

7) The review just stops at the end without a real final concluding sentence.  The conclusion section should be reorganized.

Author Response

We appreciate the reviewer’s careful reviewing and the concerns raised. We have taken a thorough effort to address all critiques.

 “1) The English grammar needs extensive editing by a native English speaker qualified in the field.  There are numerous errors, some of which are meaningful.  For example, the authors use 'master cells'  instead of 'mast cells' on lines 131, 154, and 155.  On line 151 they use 'transfection factor' instead of 'transcription factor.'”

Authors' response – We apologize for these typos. “Master” in lines 134, 158, 159, and Fig 3A has been corrected to “mast” along changing line 153 “transfection factor” to “transcription factor”. Grammatic errors have been thoroughly edited by three native speakers (Drs Gu, Kapoor, and Major). We trust manuscript’s readability has been improved.

“2) Figure 4 indicates that there is no change in RKIP transcript levels for localized prostate cancers.  It would be helpful if the authors include a similar panel and analysis for metastatic prostate cancer using SU2C or other similar datasets.  Also, are there differences in patients with mutations in KRAS?”

Authors' response – We thank the reviewer for these insights. By using the well-established Sawyers microarray dataset containing primary and metastatic PCs, panel E was added to Figure 4 to show RKIP mRNA expression was not significantly reduced in mPCs compared to primary PCs. SU2C is one of most comprehensive mPC dataset which has been updated in 2019 (Abida et al, PNAS 116, 11428-36). Although the dataset was not designed for analyzing different gene expression between primary and metastatic PCs, we have used it to show the inverse correlations of RKIP with a set of immune checkpoints in mPC (lines 429-433); the same associations in primary PCs were included in the last submission. This addition strengthens a potential role of RKIP in facilitating immune attack on PC. Collectively, analyses of RKIP expression in mPCs and its association with immune checkpoints in mPCs add to this review, for which we appreciate the reviewer’s comments.

With respect to KRAS (the most frequently mutated RAS among 3 RAS genes), it is not among the most frequently mutant genes in PC, which include SPOP, TP53, FOXA1, and PTEN (The Cancer Genome Atlas Research Network, Cell 163, 1011-25, 2015). In accordance with this notion, 2 primary PC tumors (0.4%) in the TCGA PanCancer Atlas PC cohort (n=494) contained KRAS mutations (G12R and G12D) and 4 mPCs (0.9%) in the SU2C dataset (429 patients/444 tumors) harbored KRAS mutation: G12R, G12D, Q61K, and A146V. The low mutation rate of KRAS in PC is unlikely a factor relevant to RKIP’s tumor suppressive function in PC.

“3) There seem to be some conflicts in the literature regarding whether changes in RKIP affect cellular proliferation.  Some papers say that it does and others that it does not.  The authors should comment as to why that might be, and try to resolve this or offer strengths and weaknesses of the different studies.”

Authors' response – We agree with the reviewer that differential impacts of RKIP on cellular proliferation have been reported, which might be in part attributed to RKIP inhibiting Raf1-MED-ERK and GPCR (G protein-coupled receptor) signaling when its S153 is nonphosphorylated; the phosphorylation reverses its actions on both signaling (Figure 2), two pathways well-known for their pro-oncogenesis activities. In our review, we discussed weaknesses of key studies. These weaknesses were indicated in the last submission in two forms: suggestions and perspective. In this revision, weaknesses have been discussed in a straightforwardly manner, while suggestions to indicate weaknesses remain. Also, in this revision, a single “Perspectives” section (section 5) has been organized in response to Reviewer #3 comment #7; this revised structure emphasizes weaknesses of the field (i.e. RKIP related PC research). Finally, we have touched the base for different impact of RKIP in regulating cellular proliferation in section 6: Conclusions. We hope these efforts collectively address the reviewer’s points.

“4) The authors discuss miRNAs and lncRNAs extensively, and the fact that the mRNA of RKIP doesn't change but the protein does, and yet they focus on the effects of miRNAs on transcript levels and not on potential translation efffects.”

Authors' response – We see reviewer’s points. 1) We might have overstated the possibility that RKIP downregulation does not occur at the transcription level. In this revision, we have toned down this claim (lines 226-227) by stating “… … RKIP downregulation observed in PC occurs at least in part at post-mRNA levels.” Based on the evidence of inhibiting RKIP by EMT transcription factors (like Snail), the possibility of reductions in RKIP expression via transcription regulation cannot be excluded; 2) miRNA can inhibit translation; and 3) there is lack of studies on post-translational modifications on RKIP in PC, which prevents our evaluations on this domain.

“5) The first paragraph on page 8 has a long discussion of miR-453.  It is not clear why this is relevant entirely.  Also, it is also referred to as miR-543.  It is not clear if this is a typo or two different miRNAs.”

Authors' response – Our typos. It should be miR-543; it has been corrected (lines 312, 314, and 315). We thank the reviewer for pointing out the error.

“6) The sentence on line 342 should have a reference at the end.”

Authors' response – Reference 142 has been added (line 354, this revision).

“7) The review just stops at the end without a real final concluding sentence.  The conclusion section should be reorganized.”

Authors' response – We thank the reviewer to outline this structural deficiency. In this revision, the Conclusion section (section 6) is separated from the Perspectives section (section 5). This revised setting should have solved this issue.

Round 2

Reviewer 3 Report

The manuscript is much improved.  There are still a few minor grammatical errors.

Line 364: RKIP is involved

Lines 452 & 458: RKIP downregulation

Line 479: missing 'the' (regulating the tumor microenvironment)